# Basolateral and central amygdala orchestrate how we learn whom to trust

Ronald Sladky [1,4✉], Federica Riva[1,4], Lisa Anna Rosenberger[1], Jack van Honk[2,3] & Claus Lamm [1✉]

Cooperation and mutual trust are essential in our society, yet not everybody is trustworthy. In this fMRI study, 62 healthy volunteers performed a repeated trust game, placing trust in a trustworthy or an untrustworthy player. We found that the central amygdala was active during trust behavior planning while the basolateral amygdala was active during outcome evaluation. When planning the trust behavior, central and basolateral amygdala activation was stronger for the untrustworthy player compared to the trustworthy player but only in participants who actually learned to differentiate the trustworthiness of the players. Independent of learning success, nucleus accumbens encoded whether trust was reciprocated. This suggests that learning whom to trust is not related to reward processing in the nucleus accumbens, but rather to engagement of the amygdala. Our study overcomes major empirical gaps between animal models and human neuroimaging and shows how different subnuclei of the amygdala and connected areas orchestrate learning to form different subjective trustworthiness beliefs about others and guide trust choice behavior.

[1] Social, Cognitive and Affective Neuroscience Unit, Department of Cognition, Emotion, and Methods in Psychology, University of Vienna, Vienna, Austria. [2] Department of Psychology, Utrecht University, 3584 CS Utrecht, The Netherlands. [3] Department of Psychiatry and Mental Health, MRC Unit on Risk & Resilience in Mental Disorders, University of Cape Town, Observatory, 7925 Cape Town, South Africa. [4] These authors contributed equally: Ronald Sladky, Federica Riva. ✉email: ronald.sladky@univie.ac.at; claus.lamm@univie.ac.at

Human societies are built on cooperation and mutual trust. On the individual level, trusting another person entails potential rewards, but also risks if the other person is abusing our trust to our own disadvantage. Thus, learning to distinguish the trustworthiness of an interaction partner is important for successful social interactions. Research on rodents suggests an essential role of the basolateral amygdala (BLA) in learning from social experiences[1]. In line with this, we showed in a previous study that human participants with selective bilateral BLA damage failed to adapt their trust behavior towards trustworthy vs. untrustworthy interaction partners in a repeated trust game[2]. However, functional reorganization after degenerative brain damage might prevent the generalizability of these findings to neurotypical populations. Neuroimaging in neurotypical populations indeed did not consistently report involvement of the amygdala in trust behavior. This might be explained by difficulties in differentiating between the amygdala's structurally and functionally different subnuclei, i.e., the BLA and central amygdala (CeA), which have even antagonistic features particularly in trust behavior[3].

The amygdala is widely regarded as paramount for social cognition[4], but it has been investigated as a uniform structure in the majority of human neuroimaging studies[5]. While this approach may be due to the limited spatial specificity of functional MRI particularly in the ventral brain[6,7], it ignores the structural and functional heterogeneity of this brain area and its subnuclei[8]. Here, we overcame the limitations of previous research by using an acquisition protocol optimized for imaging ventral brain areas[9] in combination with a multiband EPI sequence with high spatial and temporal resolution[10], allowing for time-resolved analysis of amygdalar subnuclei.

Our recent research in participants with BLA lesions[2] proposed that a network centered around the BLA adaptively subserves learning to trust and to distrust others. Importantly, this novel insight was based on a trust game task in which the participants repeatedly interacted with a trustworthy and an untrustworthy interaction partner. The task thus allowed us to investigate the dynamics of trust formation, as well as the role that different decision-making processes play in that. Here, using functional MRI in a healthy neurotypical population we employ the exact same behavioral paradigm to confirm and extend these findings to the specific functions of the separate subnuclei of the amygdala and the networks they are a part of.

Our aims were to derive what role the different amygdala subnuclei play for different aspects relevant in learning whom to trust, and to link them to neural activation in other subcortical regions that are highly connected with the amygdala. This means, the primary goal of our analyses was to investigate task-dependent BLA and CeA function and how BOLD response changes (a) during the different task phases that require different cognitive functions, (b) over the course of the experiment where participants learn the task with varying degrees of success, and (c) how it is affected by differences in trust behavior and subjective trustworthiness ratings. The secondary and more explorative goal of our analyses was to determine the involvement of other highly relevant subcortical brain regions[11]. More specifically, nucleus accumbens (NAc) was chosen due to its relevance in reward learning[12,13] and social decision making[14] in conjunction with two dopaminergic midbrain regions, the substantia nigra and the ventral tegmental area (SN/VTA). Additionally, the bed nucleus of the stria terminalis (BST) is considered a part of the extended amygdala complex[15,16] and could play an antagonistic role in this task, given its involvement in threat encoding mechanisms[17–19]. Finally, the septal area of the basal forebrain could be of particular relevance during trust decisions[20].

## Results

Participants played the repeated trust game inside the MRI scanner, ostensibly with what turned out to be one trustworthy and one untrustworthy player. In reality, the two players were both simulated, with their returns following a preprogrammed response schedule. Within one session and run, in total, participants played 20 rounds with each of the two players (2 × 20 rounds). In general, participants were able to adapt their trust behavior, i.e., investments in the trust game, to the trustworthy and the untrustworthy player. However, there was a marked variability within our study sample, which allowed for a partition into a *learner* and *non-learner* subgroup (Fig. 1). The task consisted of four different task phases (i.e., the *preparation, investment, waiting,* and *outcome* phase). A detailed time-resolved analysis of the BLA and CeA revealed that activation changed over the course of the different task phases. We found maximum BLA activation in the *outcome* evaluation phase and maximum CeA activation in the *preparation* phase. Yet, there was no overall BLA and CeA activation difference between the trustworthy or untrustworthy player in any of the task phases (Fig. 2). However, when differentiating between learners and non-learners, we observed more activation in the BLA and the CeA for the untrustworthy player during the *introduction* phase of a trust game round (Fig. 3). Additionally, while nucleus accumbens (NAc),

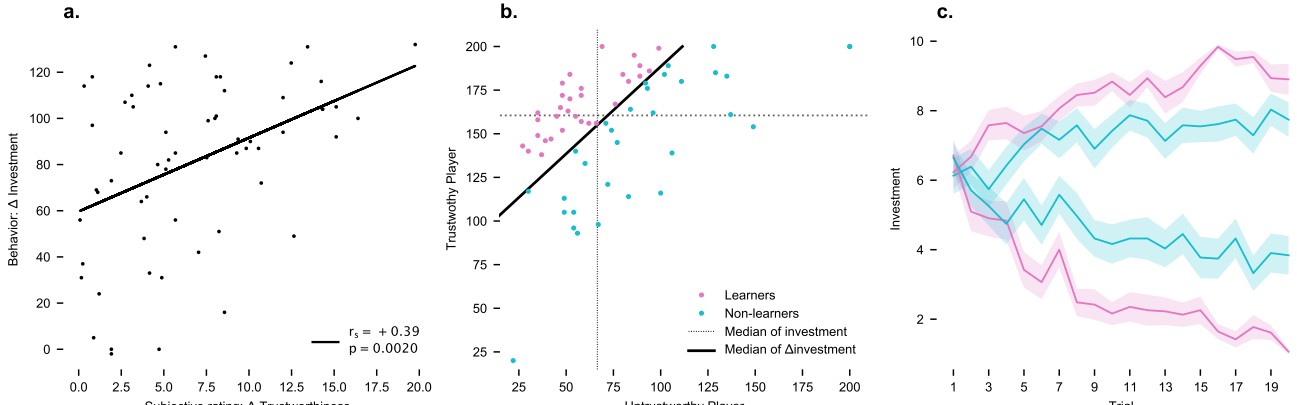

**Fig. 1 Behavioral results. a** Investment vs. trustworthiness. Behavioral trust (Δ investment) correlates with post-experiment subjective ratings (Δ trustworthiness rating), $r_s = +0.39$, $p = 0.002$. **b** Participants' investment behavior. In total, participants invested more in the trustworthy player. The difference between the investment into the trustworthy and untrustworthy player (Δ investment) was used to median-split the population into a subgroup that learned to differentiate (learners, magenta color) and those who did not (non-learners, cyan color). **c** Participants' investment behavior over time. After a few rounds, learners adapted their investment behavior to favor the trustworthy player. This differentiation was reduced in non-learners. Plot displays mean and SEM.

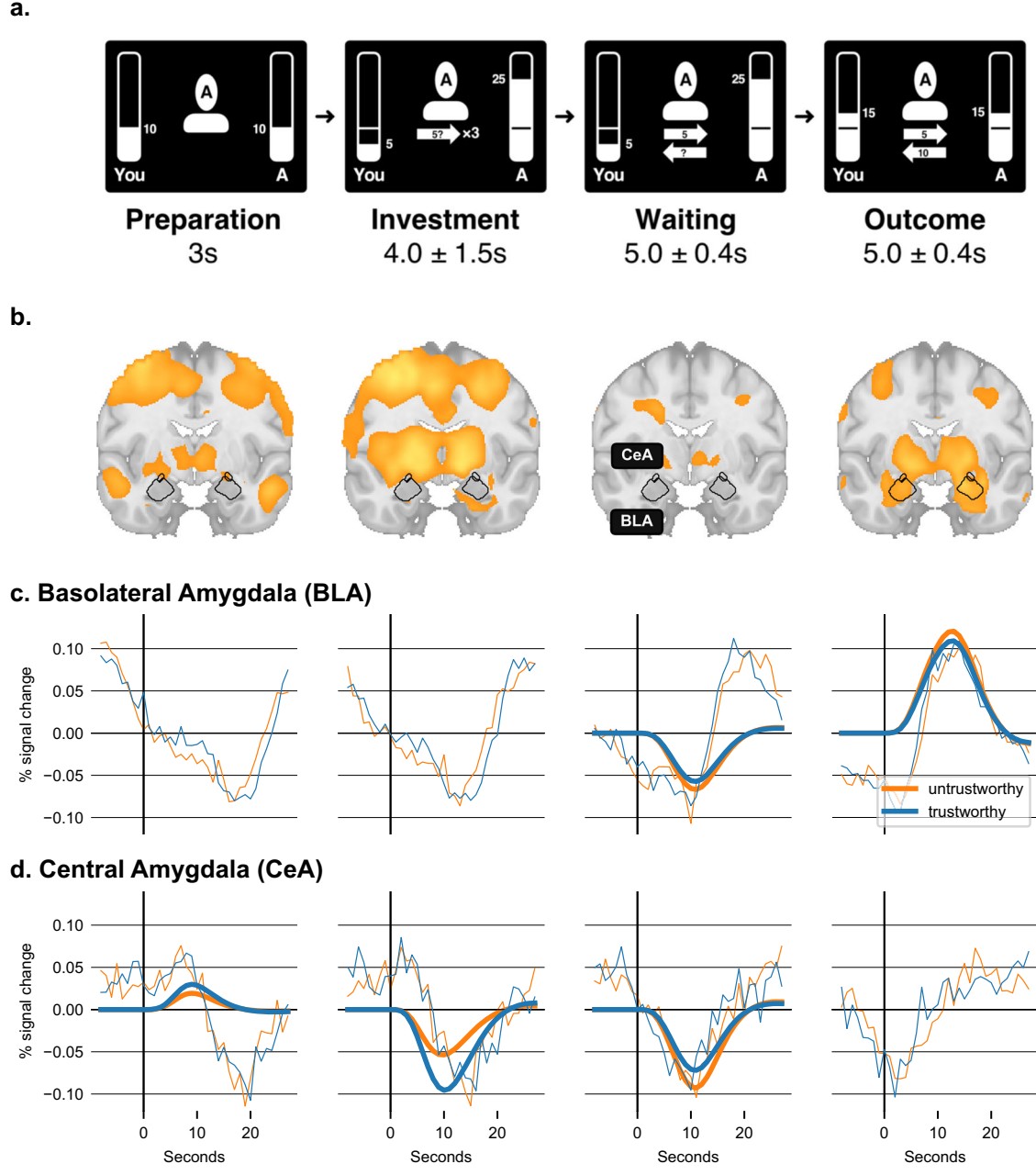

**Fig. 2 Trust game task phases and phase-dependent amygdala BOLD response. a** fMRI implementation of the trust game. Inside the MRI scanner, participants played the repeated trust game alternating with a (simulated) trustworthy and an untrustworthy player (2 × 20 rounds). *Preparation Phase.* Participants were presented with the face of the player they played with in this round. Both received an endowment of 10 points at the outset of each round. *Investment Phase.* Participants were asked to select an amount of 1 to 10 points to invest in the present player. The amount invested was tripled and added to the player's account. *Waiting Phase.* While the players made their decision, the participant needed to wait. *Outcome Phase.* Finally, the player transferred back points to the participant, resulting in a non-negative outcome for the trustworthy (as shown in the example) and a non-positive outcome for the untrustworthy player. **b** Statistical parametric maps (SPMs) and outline of the anatomically defined Volumes of Interest (VOIs) of BLA and CeA. SPMs show contrast for both players combined vs. baseline and are thresholded at p < 0.001 for display purposes. **c, d** Time course of BLA and CeA BOLD responses. CeA but not BLA was activated during the *preparation* phase, while BLA but not CeA was activated during the *outcome* phase. There were no activation differences between the trustworthy player (blue) and the untrustworthy player (orange). Thick lines represent the estimated BOLD response from the same SPM model shown in panel B (not shown if *p* > 0.05 Bonferroni corrected) and fine lines represent the actual data (average VOI time courses).

substantia nigra and ventral tegmental area (SN/VTA), and bed nucleus of the stria terminalis (BST) activity was increased for the trustworthy player during *outcome* evaluation, there was no group difference between learners and non-learners (Fig. 4).

**Behavioral results**. Marked trust differences emerged across the whole sample in the investment behavior towards the trustworthy as

opposed to the untrustworthy player, with participants generally investing more in the trustworthy player on average, and increasingly so over the course of the repeated rounds of the task (Fig. 1b & c). Morever, we find that individual differences in behavioral trust ($\Delta$ investment = investment_trustworthy − investment_untrustworthy) showed a positive correlation with subjective trustworthiness ratings ($\Delta$ trustworthiness = trustworthiness_trustworthy − trustworthiness_untrustworthy),

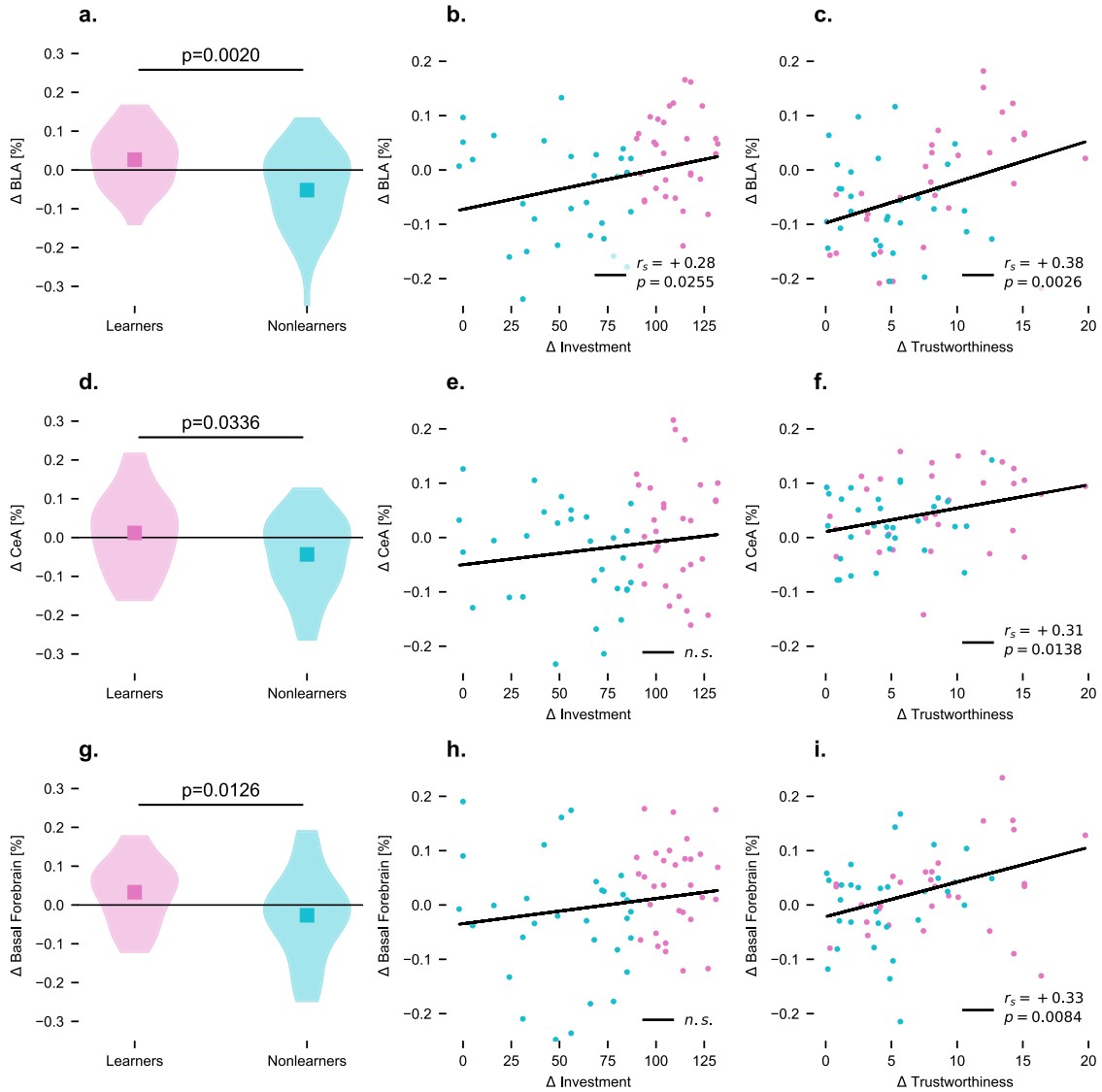

**Fig. 3 Activation differences between untrustworthy and trustworthy player in the *preparation* phase.** BLA activation differences (contrast: untrustworthy–trustworthy) were higher for learners (magenta) vs. non-learners (cyan) (**a**), correlated with investment differences (**b**) and post-experiment subjective trustworthiness rating differences (**c**). The same relationship was found for CeA (**d**, **f**), except the correlation with investment differences was not significant (**e**). An exploratory analysis of the basal forebrain showed a similar relationship (**g–i**).

$r_s = +0.39$, $p = 0.002$ (Fig. 1a). On the subjective level, the trustworthy player was rated as significantly more trustworthy, fairer, and more intelligent than the untrustworthy player (all $p < 0.05$, Bonferroni corrected), but not as more attractive (n.s., after Bonferroni correction).

**Neuroimaging results**. We find that different subnuclei of the amygdala engaged in the trust game show increased activation during different phases of the task paradigm. This suggests that they are supposedly related to different aspects and processes required by the formation of trust. The two subnuclei that played the most specific role (Fig. 2b) were the BLA and the CeA. Notably, the activation differences in these subnuclei and the validity of our analysis approach are supported by differences in their functional connectivity profiles, determined in our data. While the BLA connected to sensory integration areas and lateral PFC, the CeA connected to the ventral striatum, including the nucleus accumbens, and areas in the medial PFC (Supplementary

Fig. 2). The role of these subnuclei in the different task phases is as follows.

In the BLA, there was no significant (at $p < 0.05$ Bonferroni-corrected) activation during the *introduction* (PSC ± 95% CI = $-0.00 \pm 0.03$, $t_{61} = 0.2$, $p = 0.8530$, $d = 0.0$) and the *investment* phases (PSC ± 95% CI = $-0.04 \pm 0.04$, $t_{61} = 2.2$, $p = 0.0293$, $d = 0.3$), but significant deactivation during the *waiting* (PSC ± 95% CI = $-0.07 \pm 0.03$, $t_{61} = 5.2$, $p < 0.0001$, $d = 0.7$) and significant activation during the *outcome* phases (PSC ± 95% CI = $+0.11 \pm 0.02$, $t_{61} = 9.8$, $p < 0.0001$, $d = 1.3$) (Fig. 2c). In the CeA, there was significant activation during the *introduction* phase (PSC ± 95% CI = $+0.04 \pm 0.02$, $t_{61} = 3.2$, $p = 0.0021$, $d = 0.4$), significant deactivation during the *investment* (PSC ± 95% CI = $-0.12 \pm 0.03$, $t_{61} = 6.6$, $p < 0.0001$, $d = 0.8$) and *waiting* phases (PSC ± 95% CI = $-0.10 \pm 0.03$, $t_{61} = 6.8$, $p < 0.0001$, $d = 0.9$) but no significant effect during the *outcome* phase (PSC ± 95% CI = $-0.01 \pm 0.02$, $t_{61} = 1.1$, $p = 0.2972$, $d = 0.1$) (Fig. 2d). All $p$-values reported as significant survive Bonferroni correction for multiple comparisons (at $p < 0.05$ Bonferroni- corrected).

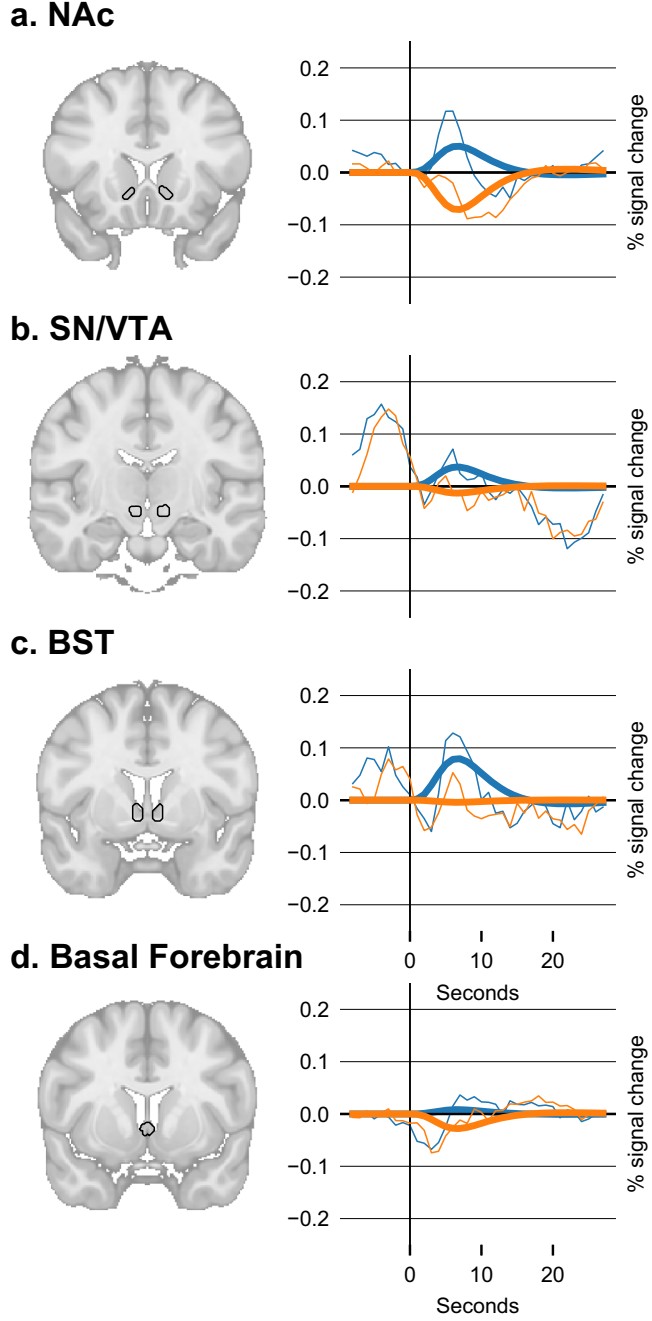

**a. NAc**

**b. SN/VTA**

**c. BST**

**d. Basal Forebrain**

**Fig. 4 More activity for the trustworthy (blue) vs. untrustworthy player (orange) during the *outcome* event found in the secondary analyses. a** In the nucleus accumbens (NAc), $t = +6.3$, $p < 0.0001$, **b** Substantia nigra (SN) and ventral tegmental area (VTA), $t = +3.7$, $p = 0.0005$, **c** Bed nucleus of the stria terminalis (BST), $t = +4.00$, $p = 0.0002$, and (**d**) the septal part of the basal forebrain, $t = 2.1$, $p = 0.0361$. Thick lines represent the estimated BOLD model.

Note that these response patterns were irrespective of whether a participant played with a trustworthy or untrustworthy player, as there were no significant differences between these two conditions. These findings thus relate to the general role of the amygdala subnuclei in the different parts of the task, and the overall processes and subfunctions engaged by the trust decision.

As a next step, we aimed to pinpoint how the engagement of the amygdala was related to differential evaluations of trustworthiness, and the resulting trust behavior toward the two players. Individual difference analyses showed a relationship between BLA and the CeA activation in the *preparation* phase and subjective trustworthiness and behavioral trust measures. More specifically, we first used a median split of $\Delta$ *investment* (Fig. 1b) to distinguish learners from non-learners (i.e., those who adjusted their investment behavior less to the trustworthiness of the player), and then assessed how they differed in their amygdala activations. A Mann–Whitney $U$-test showed that during the *preparation phase*, the activation difference between untrustworthy–trustworthy player was significantly larger in learners than in non-learners in the BLA ($\Delta$ PSC ± 95% CI = +0.08 ± 0.03, U = 686, $p = 0.002$, $d = 0.8$, Fig. 3a) and in the CeA ($\Delta$ PSC ± 95% CI = +0.06 ± 0.02, U = 611, $p = 0.0336$, $d = 0.5$, Fig. 3d). Moreover, the BLA activation differences between untrustworthy vs. trustworthy players in this phase correlated positively with behavioral trust ($\Delta$ *investment*), $r_s = +0.28$, $p = 0.0255$ (Fig. 3b) and subjective trustworthiness ($\Delta$ *trustworthiness*), $r_s = +0.38$, $p = 0.0026$ (Fig. 3c). CeA activation differences correlated with subjective trustworthiness ratings, $\Delta$ *trustworthiness*, $r_s = +0.31$, $p = 0.0138$ (Fig. 3f), but not with behavioral trust, $\Delta$ *investment* (Fig. 3e). While considering whether to trust or distrust a player, the CeA in learners thus seems primarily linked to evaluations of trustworthiness, whereas the BLA is additionally relevant for the actual behavioral outcome as well as whether someone efficiently learns to adapt behavior to the actually reciprocated trust or not. Moreover, these relationships are driven by stronger engagement for rounds with the untrustworthy (compared to the trustworthy) player, suggesting that what is coded is rather the absence than the presence of trust. Finally, an exploratory analysis of the forebrain showed a similar activation difference between the trustworthy and untrustworthy player for learners vs. non-learners ($\Delta$ PSC ± 95% CI = + 0.06 ± 0.02, U = 640, $p = 0.0126$, $d = 0.6$, Fig. 3g) that, like the CeA, correlated with $\Delta$ *trustworthiness*, $r_s = +0.33$, $p = 0.0084$ (Fig. 3i) but not $\Delta$ *investment* (Fig. 3h).

The neural responses in the *preparation* phase mainly provide insights into how the acquired information about a player's trustworthiness drives the decisions of participants. The activation in the *outcome* evaluation phase, on the other hand, tells us about how this information is acquired and possibly updated. As outlined above, we observed overall activation in the BLA during the *outcome* evaluation phase (Fig. 2), and this may be linked to reward processing[21]. Surprisingly, though, we did not find differences between the trustworthy and untrustworthy player in the BLA or CeA in the outcome phase, and neither did we find correlations with trust behavior and trustworthiness rating. We thus extended our analyses to subcortical regions with particularly strong anatomical and functional connections to the amygdala. These were the nucleus accumbens (NAc), as well as the dopaminergic midbrain, comprising substantia nigra and the ventral tegmental area (SN/VTA), relevant for encoding reward, and the bed nucleus of the stria terminalis (BST), relevant for encoding threat[17–19].

When the *outcome* of the player decision was presented, higher activations were observed for the trustworthy compared to the untrustworthy player in the NAc ($\Delta$ PSC ± 95% CI = +0.11 ± 0.03, $t_{61} = 6.3$, $p < 0.0001$, $d = 0.8$), SN/VTA ($\Delta$ PSC ± 95% CI = +0.05 ± 0.03, $t_{61} = 3.7$, $p = 0.0005$, $d = 0.5$), BST ($\Delta$ PSC ± 95% CI = +0.08 ± 0.04, $t_{61} = 4.0$, $p = 0.0002$, $d = 0.5$), and the basal forebrain ($\Delta$ PSC ± 95% CI = +0.03 ± 0.03, $t_{61} = 2.1$, $p = 0.0361$, $d = 0.3$) (Fig. 4). Moreover, the gain or loss (i.e., *back-transfer - investment amount*) correlated with NAc ($r_s = +0.19$, $p < 0.0001$) and BST ($r_s = +0.10$, $p < 0.0001$), but this was irrespective of the activation difference between trustworthy and untrustworthy player.

## Discussion

Our previous study in BLA-damaged participants highlighted that the BLA is indispensable for learning to differentiate between trustworthy and untrustworthy players in the trust game[2]. This has important implications for our understanding of social decision-making in humans and, most likely, other mammals[22]. However, extending these findings to the neural networks connected to the amygdala in healthy, neurotypical, human participants is of the essence. Here we confirmed the relevance of the BLA for distinguishing between trustworthy and untrustworthy players based on previous experience and how, in conjunction with the CeA, it plays a role in the guiding of trust behavior. Specifically, BLA activity was increased during the processing of the outcome of the player's behavior but unselectively for trustworthy vs. untrustworthy player. Instead, we found increased activation in the NAc, BST, and SN/VTA for the trustworthy vs. untrustworthy player during outcome processing. Importantly, here we did not observe an activation difference between learners and non-learners. This could indicate that learners and non-learners processed the outcome in a similar fashion, suggesting that their understanding of the task and motivation were comparable. This further highlights the central role of the BLA for trust learning.

Indeed, we found the BLA to be most active during outcome evaluation, i.e., when participants learned whether their trust was reciprocated or not, suggesting that it plays an important role in acquiring beliefs about the trustworthiness of others. It appears, however, that the BLA was not directly involved in building specific outcome expectations during the waiting and evaluation phase. The BOLD response in the BLA was not modulated by the trustworthiness or the players' back-transfer amount, unlike activity in the NAc, SN, and BST. This highlights that the BLA, although indispensable for learning whom to trust[2], as indicated by our previous research, is only a component of a complex brain network for reward processing and social evaluation.

In addition, we found that while participants prepared for their next investment, the BLA together with the CeA exhibited increased activation for the untrustworthy player. Importantly, this activation difference was only found in those participants who learned to differentiate between the players, indicating its role in (1) guiding trust behavior as BLA activation differences directly precede the participant's investment behavior and also (2) in trustworthiness evaluation, as BLA and CeA BOLD responses correlated with the subjective rating after the experiment.

Nowadays, it is a well-established finding that a sub-population of BLA's neurons selectively respond to reward, whereas other sub-populations either only respond to aversive stimuli[23], or selectively increase their firing rate when the rewarding or aversive stimulus was unexpected, i.e., not predicted[24] (which means that something novel has to be learned about the environment). In the context of our findings, this view supports the notion that the BLA is relevant for encoding both the rewarding behavior of the trustworthy player and the aversive behavior of the untrustworthy player. Additionally, we can speculate that optimal performance in the trust game does not only rely on reward learning and threat detection, but also on predicting affective consequences based on abstract information. Supporting evidence for this theory can be found in a recent study in a patient with acquired complete bilateral amygdala lesions (patient SM, 49 years old, female), who showed impairments in making good predictions about what kind of written statements will induce fear[25].

The fact that we did not observe any habituation in any of the amygdala subregions (Supplementary Fig. 3) indicates that the BLA not only responds to novel stimuli but is relevant for the continuous encoding and updating of information of social experiences. In the light of the recent debate on amygdala BOLD signal habituation[26–30] this finding could be important for the development of additional tasks that robustly activate the amygdala.

While BLA's activation during outcome evaluation suggests its involvement in discriminating and tracking outcome-specific effects, the CeA is involved in general motivational aspects of reward-related events[31] and, thus, might not play a role in the actual learning process in the outcome phase. Instead, we found it active during the preparation phase, which immediately preceded the investment phase. This could indicate that the CeA was regulated by the BLA output, which has been demonstrated before for a different task in a cross-species model[32]. As CeA activity was increased before the participant's investment, it might play a role in controlling trust behavior. More importantly, CeA activity during the preparation phase correlated with the subjective rating of trustworthiness of the player, indicating that it could be relevant for encoding the affective value attached to the player.

During outcome evaluation, we observed increased activation in the bed nucleus of the stria terminalis (BST), which, together with the CeA, is considered the extended amygdala complex[15,16]. The BST has been suggested to play a role in both reward processing and social cognition[33] and exhibits strong connections to the NAc[34]. While the CeA is associated with fast fear responses (e.g., startle reflex), the BST is responsible for slower effective learning processes[35] and has been linked to adaptive and maladaptive responses to sustained stress and threat[17,36]. Of note, the BST plays a particular role in dealing with unpredictable threat[37], which could be the case in an uncertain social investment. However, these two views are still part of ongoing debates[38,39]. Most recently, the BST was shown to be more involved in fear-related anticipation processes, whereas the CeA was linked to threat confrontation[19]. In this study, we found the BST to be involved in the outcome evaluation phase. Based on the literature, it could be expected that the BST would show more activation for the aversive untrustworthy player, which was not the case. Instead, we observed that the BOLD responses of BST and NAc were both more activated by the trustworthy player. The NAc and other striatal areas are known to be involved in evaluating the players trustworthiness based on their back-transfer behavior[40–42] and amygdala to NAc coactivation are relevant for social decision making[14]. Rodent research has shown that BLA to NAc connections mediate reward learning[12,13]. Importantly, stimulus-evoked excitation of NAc neurons depends on input from the BLA and is required for dopamine to enhance the stimulus-evoked firing of NAc neurons, ultimately, leading to reward-seeking behavior[43]. This could mean that both regions might engage in a synergetic fashion, where the NAc would be particularly relevant for tracking rewards. The BST, on the other hand, could be responsible for increasing arousal as generous investments in the trustworthy player also entail a potential threat of betrayal. These findings suggest a functional dissociation between reward and risk evaluation based on the observed outcome of one's behavior, which appeared to be comparable in non-learners, and the mechanisms of trust learning. Finally, the observed difference in the septal area of the basal forebrain during outcome evaluation could indicate its involvement in updating beliefs about the other players' trustworthiness. This would be in line with previous research that suggested that this region encodes the participant's volatility beliefs (i.e., an estimator for expected uncertainty) in both social and non-social learning tasks[44,45]. In the context of our study, we also observed an involvement during the introduction phase, showing more activation in learners relative to non-learners for the untrustworthy relative to the

*trustworthy* player. It has been shown before that the basal forebrain is involved in trust decisions[20] and its differential activity during *outcome* evaluation (higher for the *trustworthy* player) and the *introduction* phase (higher for the *untrustworthy* player in learners) highlights its potentially crucial role in trust learning.

In sum, we confirm that the BLA was indeed involved in learning whom to trust and that observations from amygdala-lesioned participants can be translated to healthy neurotypical participants. Additionally, our fine-grained, time-resolved analyses of the amygdala subnuclei and the functionally-connected brain areas provided important insights into different cognitive mechanisms involved in trust learning. We found that the BLA was relevant for *discriminating* between trustworthy and untrustworthy players based on previous experience and for *optimizing trust behavior*. Only in those participants who learned to optimize their investments, we found selectively more activation in the BLA during the planning of a new investment that required trust. The BLA was also active during outcome evaluation suggesting its involvement in the process of *belief formation* based on the players' back-transfer amount. As we did not observe a difference between the trustworthy or untrustworthy player, we can assume that encoding of potential rewards and risks is mediated by the NAc and BST, respectively, which showed a selectively increased activity for the trustworthy player or an increased investment. Finally, the CeA is known to receive inputs from the BLA and BST, and exhibited the largest BOLD response during the *planning* phase. CeA activity did not correlate with the participant's trust behavior, however, there was a correlation with the participant's subjective belief of the players' trustworthiness. This suggests that the CeA could encode subjective value, possibly also indirectly affecting trust behavior via the BLA. Taken together, our work suggests that there is a high demand for translational work on the amygdala, its subnuclei, and connected brain regions. Based on the present results, we propose that careful variations of the trust game in combination with computational modeling may serve as an experimental model to further uncover the neural mechanisms underlying human social cognition and behavior.

## Materials and methods

**Participants**. Sixty two healthy, neurotypical volunteers (age = 23.83 ± 3.15 years, *f/m* = 31/31), mostly undergraduate students from Vienna, Austria were recruited. Exclusion criteria were standard MRI exclusion criteria (e.g.: pregnancy, claustrophobia, and MRI-incompatible implants, clinically significant somatic diseases), a history of psychiatric or neurological disorders, substance abuse, psychopharmacological medication, less than nine years of education, as well as not being task-naive (e.g., having already participated in a similar study or being a psychology student). All participants provided written informed consent in accordance with the Declaration of Helsinki and were compensated for their participation. The study was approved by the ethics committee of the Medical University of Vienna (EK-Nr. 1489/2015).

**Procedure and task**. This study was part of a bigger project including two additional tasks and a sample of older adults, which are not reported in the current article. Participants were first invited to a screening session where they performed some cognitive tasks and filled in some self-reported measures of psychological traits. The main session was usually conducted within two weeks from the screening session. Participants were welcomed to the MRI facility (University of Vienna MR Center) together with two other participants, who were in fact two confederates of the experimenter invited to play the players' role. After having signed the consent form and filled in the MR safety questionnaire, participants and confederates were introduced to the protocol of the whole session. Afterwards, they went through the training of the three tasks, including the trust game. At the end of the training, participants were required to answer some questions in order to make sure they understood the task. Participants were finally placed into the MR scanner, while the confederates were putatively playing the task in the computer room next to the scanner room.

The repeated trust game was adapted from our previous study[2] and programmed in z-Tree (version 3.3.7[46]). The script of this trust game is deposited online[2]. In short, two players per round, an investor and a player, exchange

monetary units with the aim to maximize their monetary outcome. In total, 40 rounds were played and the participant always played the role of the investor, while the players were allegedly played by the two confederates in an alternate randomized order. In reality, the actions taken by the two players were preprogrammed in a way that one of the confederates was behaving in a trustworthy and the other one in an untrustworthy way. Confederates/players were of similar age and the same gender as the participant. At the beginning of each round (i.e., 20 per trustworthy condition and 20 per untrustworthy condition) both players received an endowment of 10 monetary units. Then each round encompasses four phases. In the *preparation* phase, participants are presented with the picture of the player's face they are playing with in the current round. In the *investment* phase, participants invest (part of) their endowment (at least 1 unit) and the investment is tripled and then transferred to the player. During the *waiting* phase, the players ostensibly perform their back-transfers. Finally, during the *outcome* phase, participants are presented with the back-transfer outcome. In the first two rounds, both the trustworthy and untrustworthy players back-transferred the same amount of the money invested to the participants. In the following rounds, the trustworthy player always back-transferred as much or more than the money invested by the player, whereas the untrustworthy player always back-transferred less than or as much as the money invested by the investor. The sums invested by the participants were considered as a measure of trust given to the two players by the participants and used as the main variable of interest. Points earned throughout the task were transformed to Euros and added to the participants' compensation.

At the end of the task, participants were presented with the players' picture and were asked to rate them on four adjectives: trustworthiness, fairness, attractiveness, and intelligence (original German: *Wie vertrauenswürdig/attraktiv/intelligent/fair haben Sie den/die Teilnehmer/in wahrgenommen?*). Ratings were provided on visual analog scales and transformed off-line to a numerical range between −10 and +10.

**Behavioral data analysis**. It is commonly understood that participants' investment behavior is a behavioral expression of how they judged the players' trustworthiness and changes reflect the extent to which they updated their beliefs[2,47,48]. This *objective* measure of trust was used to distinguish between learners and non-learners (using the median as cut-off value) and for a Spearman correlation analysis between the subjective ratings (trustworthiness, fairness, attractiveness, and intelligence) and the BOLD response in the amygdala.

**Functional MRI Data acquisition, processing, and analyses**. MRI acquisitions were performed on a Skyra 3 Tesla MRI scanner (Siemens Healthineers, Erlangen, Germany) using the manufacturer's 32 channel head coil at the MR Center of the University of Vienna. In a single session, one run of the repeated trust game was performed by the participant while we performed functional MRI using a gradient echo T2*-weighted echo planar image sequence with the following parameters: MB-EPI factor = 4, TR/TE = 704/34 ms, $2.2 \times 2.2 \times 3.5$ mm$^3$, $96 \times 92 \times 32$ voxels, flip angle = 50°, $n < 2400$ volumes.

Data processing and analyses of the functional MRI data were performed in SPM (SPM12, http://www.fil.ion.ucl.ac.uk/spm/software/spm12/) and the Python projects nipype (http://nipy.org/nipype) and nilearn (http://nilearn.github.io). Preprocessing comprised slice-timing correction[49] using SPM, realignment using SPM, non-linear normalization of the EPI images[50] to a study-specific group template using ANTs[51], and spatial smoothing with a 6 mm FWHM Gaussian kernel using SPM. SPM result maps were warped from study space to MNI space (final resolution = $1.5 \times 1.5 \times 1.5$ mm$^3$) using ANTs. VOI analyses were performed in study space using anatomical masks that were transformed to the study-specific group template using the inverse transformation from MNI using ANTs' *ApplyTransforms*. To verify that the EPI images properly covered the VOI, axial slices of the median single-subject mean volumes in study space are presented in the supplementary material (Supplementary Fig. 1). Additionally, functional connectivity analyses using the CeA and BLA seeds were performed to validate that the time courses of these small regions can be functionally differentiated (Supplementary Fig. 2).

First-level analyses of the data were implemented using nipype and performed using SPM12's GLM approach. The GLM design matrix encompassed individual regressors for each of the 4 task phases (i.e., preparation, investment, waiting, and outcome) and each of the 2 interaction partners (trustworthy and untrustworthy, resulting in 8 effects of interest). Additionally, 6 realignment parameters were added as nuisance regressors to account for residual head motion effects. Second-level analyses of the data were implemented using nipype and performed using SPM12's group-level approach for visual inspection of the whole brain results.

Volume of interest analyses was performed on the mean timeseries extracted using nilearn's *fit_transform* from anatomical masks from the BLA, CeA[52], NAc (AAL Atlas), BST[53], SN/VTA (Talairach atlas transformed to MNI space), and basal forebrain (Jülich Brain MPM atlas). To investigate phase-dependent activation in the amygdala subregions, timeseries analyses were conducted based on the estimated percent signal change, using custom python scripts that reproduced SPM's default GLM analysis, using SPM's canonical HRF to convolve the 8 regressors of interest (4 phases × 2 players), using the realignment parameters as confounds and a DCT-based high-pass filter with SPM's default $f = 1/128$ Hz cut-off frequency to account for signal drifts. Results of the BLA and CeA VOI

analyses were Bonferrroni-corrected. Comparisons between learners and non-learners were performed using two-sampled *t*-tests and Spearman correlations of investments (i.e., trust behavior) and subjective trustworthiness ratings. Habituation was investigated using a different model, where each phase and each round was modeled individually (Supplementary Material). Using paired *t*-tests we investigated the differences in the BOLD response between the trustworthy and untrustworthy player in the other volumes of interest based on the percent signal changes as estimated by the GLM. In these secondary, exploratory analyses, no correction for multiple comparison was used. However, note that these regions, except the basal forebrain, would survive conservative Bonferroni correction.

An additional functional connectivity analysis was performed to verify that the specificity of the BOLD signal is sufficient for distinguishing between the BLA and CeA. To this end, task fMRI data were corrected for white matter and CSF signal and task effects[54] using regression before estimation of the functional connectivity maps of the BLA and CeA seeds.

**Statistics and reproducibility**. In sum, the sample of this study consisted of 62 healthy, neurotypical volunteers (age = 23.83 ± 3.15 years, *f/m* = 31/31). Data were processed and analyzed using SPM (SPM12, http://www.fil.ion.ucl.ac.uk/spm/software/spm12/) and the Python projects nipype (http://nipy.org/nipype) and nilearn (http://nilearn.github.io). Additional statistical analyses were performed using numpy and scipy; for visualization matplotlib and nilearn were used. Code and data required to reproduce our results and figures are publicly available on https://github.com/scanunit.

**Reporting summary**. Further information on research design is available in the Nature Research Reporting Summary linked to this article.

## Data availability

The data needed to reproduce the results and figures are published on our lab's public github page (https://github.com/scanunit).

## Code availability

The code needed to reproduce the results and figures are published on our lab's public github page (https://github.com/scanunit).

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

## Acknowledgements

The authors thank Helena Hartmann for her help in collecting the data. This work was funded by the Austrian Science Fund FWF (P29150). Claus Lamm and Lisa Rosenberger also acknowledge partial funding from the Vienna Science and Technology Fund (WWTF VRG13-007).

## Author contributions

Conceptualization and methodology, R.S., F.R., L.R., J.v.H., C.L.; Investigation, F.R.; Formal analysis, R.S., F.R.; Writing—original draft, R.S., F.R., L.R., J.v.H., C.L.; Writing—review and editing, R.S., F.R., L.R., J.v.H., C.L.; Funding acquisition, C.L.

## Competing interests

The authors declare no competing interests.
