## [Transparent Peer Review File · Communications Biology]

Reviewers' comments:

Reviewer #1 (Remarks to the Author):

The authors present a potentially interesting study meant to investigate more in detail the role of the different subnuclei of the amygdala and connected regions in learning whether or not to trust someone.

The authors collected a nice group size dataset, with a well studied paradigm, and present clear hypothesis informed by their previous patient study and animal literature.

The main problem I have with the current of the manuscript is that a complete, detailed and clear description of the methods is basically missing, as the information provided is surprisingly superficial.

Here below I give some examples:

- authors mentioned to have collected their data using an acquisition protocol optimized for imaging ventral brain regions, from the method though I don't see anything special except the use of multi band, which despite increasing sensitivity, is not specific to isolate the amygdala subnuclei. Additionally, from the introduction and the general difficulty to isolate the amygdala nuclei, I was expecting the authors to use much more sophisticated ways to identify the different subparts, for instance combining anatomical landmark to functional data at an individual level. The ROIs seem instead being generated by the use of normalized atlases, which are known to help with a general approximation localization of the structure of interest, and not to have the level of specificity necessary for what the authors aimed at. Anyhow, very little information is present on how the different subnuclei are identified which makes it hard to judge the quality of the specificity. Except for the small images of Figure 2b, there is nothing convincing the authors have overcome the limitation present in the literature as they claim in the introduction.
- Other nuclei beside the amygdala have been investigated. Again more information on how these nuclei have been chosen is missing. The authors very briefly mentioned a connectivity analysis, but it is not clear how this was related with the choice of these nuclei, and even more importantly how this analysis was performed
- was any correction for multiple ROI testing performed?
- authors mention to have run a whole brain analysis, but then results are not reported in the supplemental material. What the authors want to visually inspect?
- authors often report a lack of significance, I would recommend the use of Bayesian statistic before their interpretation
- authors mention that each phase of the task was modelled with a separate predictor. Considering how close in time these task phases are, I would like to see how much correlated these predictors are and how much variance is left to be explained by the different phases
- there is too little information to evaluate the ROI analysis: some text and figure suggest that a t-test per time point was calculated (and if this is the case they need to explain and show the clusters or point that are significant with appropriate correction), other part of the text suggest they just average the signal per task-phase
- authors collected measures of trustworthiness, fairness, attractiveness and intelligence. Did they then run multiple regressions? simple correlations for some of the measures, or what exactly?
- in the description of the task, authors seem to use "round" a bit differently as in one case using it with the meaning of "trial" and in another case as "session", this makes it a bit confusing to understand whether the two players were presented in separate sessions and then the order of the session were randomized across participants, or whether rounds with a different player were included in each session and therefore the order of the players randomized also within subjects.

Reviewer #2 (Remarks to the Author):

This is a very interesting and novel study, driven by clear hypotheses about the differential role of amygdala subregions in trust, which were derived from the animal literature and human lesion studies.

It is laudable that the authors used optimised fMRI sequences allowing them to disentangle different subregions of the amygdala.

Overall the methods are sound and the paper is excellently written.

Please see my more detailed comments for further clarification and thought.

Conceptual issues:

Krueger et al 2007 used optimised fMRI sequences in a trust game to examine related questions as here and found the septal region to be associated with unconditional trust. Since that paper's publication, there is been accumulating evidence on sept-hypothalamic and septal region activations being relevant for attachment-related functions, and it would be very interesting how this often forgotten part of the basal forebrain behaves in the context of the authors' task, would the authors be able to discuss this or add a supplementary analysis?

Statistical issues:

The focus on hypothesis-driven region of interest analyses is laudable and the effects appear robust, I wonder whether it would be important though to divide the results and methods sections into primary and secondary analyses more clearly to justify that there was no multiple comparison correction for all regions applied? As the main hypothesis of the paper are around two amygdala regions, I think there is no need to adjust them for multiple comparisons, but it would be helpful to clarify the hypothesis-driven, vs more exploratory 2ndary aspect.

Minor wording issues:

-The abstract: line 2: "When planning trust behavior," - insert "the" here please

-abstract line 32: replace "believes" with "beliefs"

-I was wondering whether replacing "trustee" with "player" makes more sense in that "trustee" evokes an association of being on the board of a charity or being a trusted person, but of course that is the authors' decision if they think that this is commonly used in other papers, but I find this a bit confusing for the general reader.

-The results section should use the past tense consistently

Reviewer #1 (Remarks to the Author):

The authors present a potentially interesting study meant to investigate more in detail the role of the different subnuclei of the amygdala and connected regions in learning whether or not to trust someone.

The authors collected a nice group size dataset, with a well studied paradigm, and present clear hypothesis informed by their previous patient study and animal literature.

The main problem I have with the current of the manuscript is that a complete, detailed and clear description of the methods is basically missing, as the information provided is surprisingly superficial.

We would like to thank the reviewer for thoroughly reading and commenting on our manuscript and the overall positive evaluation. Of course, we are more than glad to provide all the requested methodological details and revised the manuscript according to the reviewer's suggestions.

Here below I give some examples:

- authors mentioned to have collected their data using an acquisition protocol optimized for imaging ventral brain regions, from the method though I don't see anything special expect the use of multi band, which despite increasing sensitivity, is not specific to isolate the amygdala subnuclei. Additionally, from the introduction and the general difficulty to isolate the amygdala nuclei, I was expecting the authors to use much more sophisticated ways to identify the different subparts, for instance combining anatomical landmark to functional data at an individual level. The ROIs seem instead being generated by the use of normalized atlases, which are known to help with a general approximation localization of the structure of interest, and not to have the level of specificity necessary for what the authors aimed at. Anyhow, very little information is present on how the different subnuclei are identified which makes it hard to judge the quality of the specificity. Except for the small images of Figure 2b, there is nothing convincing the authors have overcome the limitation present in the literature as they claim in the introduction.

The reviewer addresses the important issue that subcortical and ventro-frontal brain areas are notoriously hard to investigate using susceptibility-sensitive imaging sequences as used in BOLD fMRI. Strong susceptibility-related field inhomogeneities

cause signal dropout through rapid dephasing of the spins within a voxel. Therefore, we appreciate the reviewer's concern and have expanded the details of our data processing methods and present additional results in our supplementary material. We now state that (1) we used EPI-based non-linear normalization to a study template (a method we also used in <https://www.sciencedirect.com/science/article/pii/S1053811916307467>), (2) VOI masks were transformed to the study space before single subject timeseries extraction, (3) visual inspection of EPI images demonstrates robust sensitivity in the amygdala and the other VOIs, and (4) differences in functional connectivity of BLA and CeA seeds demonstrates specificity of the BOLD signal. In detail, we changed the following sections in our manuscript:

Revised methods section (ln. 618):

Data processing and analyses of the functional MRI data were performed in SPM (SPM12, <http://www.fil.ion.ucl.ac.uk/spm/software/spm12/>) and the Python projects nipy (<http://nipy.org/nipy>) and Nilearn (<http://nilearn.github.io>). Preprocessing comprised slice-timing correction (Sladky et al., 2011) using SPM, realignment using SPM, non-linear normalization of the EPI images (Calhoun et al., 2017) to a study-specific group template using ANTs (Avants et al., 2011), and spatial smoothing with a 6 mm FWHM Gaussian kernel using SPM. SPM result maps were warped from study space to MNI space (final resolution = $1.5 \times 1.5 \times 1.5 \text{ mm}^3$) using ANTs. VOI analyses were performed in study space using anatomical masks that were transformed to the study-specific group template using the inverse transformation from MNI using ANTs' *ApplyTransforms*. To verify that the EPI images properly covered the VOI, axial slices of the median single-subject mean volumes in study space are presented in the supplementary material (**Supplementary Figure 1**). Additionally, functional connectivity analyses using the CeA and BLA seeds were performed to validate that the time courses of these small regions can be functionally differentiated (**Supplementary Figure 2**).

Revised Supplement:

VALIDATION OF FUNCTIONAL MRI DATA QUALITY

Visual inspection of the EPI images demonstrates sufficient data quality to extract time series from the volumes of interest.

Supplementary Figure 1. Median image of single-subject mean EPI volumes in study space with overlay of the VOI masks (BLA, CeA, NAc, SN/VTA, BST, Basal Forebrain). Instead of grayscale, Matplotlib's *nipy_spectral* colormap is used for better visualization of the signal intensities.

BLA and CeA functional connectivity analysis. Differences in functional connectivity when using the BLA and CeA as seed regions indicate (a) differences in functional brain network and (b) sufficient specificity in the fMRI data to robustly detect functional differences in these subnuclei.

Supplementary Figure 2. Differences in functional connectivity of BLA>CeA (hot) and CeA>BLA (cool).

- Other nuclei beside the amygdala have been investigated. Again more information on how these nuclei have been chosen is missing. The authors very briefly mentioned a connectivity analysis, but it is not clear how this was related with the choice of these nuclei, and even more importantly how this analysis was performed

We thank the reviewer for highlighting that the description of our analysis strategy was not sufficiently explicit and transparent. We added the following paragraph at the end of the introduction (ln. 68) to make it clearer that the primary analysis focused on the amygdala subnuclei, while the other regions were motivated by the literature and more exploratory in nature.

Our aims were to derive what role the different subnuclei of the amygdala play for different aspects relevant in learning whom to trust, and to link them to neural

activation in other subcortical regions that are highly connected with the amygdala. This means, the primary goal of our analyses was to investigate task-dependent BLA and CeA function and how BOLD response changes (a) during the different task-phases that require different cognitive functions, (b) over the course of the experiment where participants learn the task with varying degrees of success, and (c) how it is affected by differences in trust behavior and subjective trustworthiness ratings. The secondary and more explorative goal of our analyses was to determine the involvement of other highly relevant subcortical brain regions (Janak and Tye, 2015). More specifically, nucleus accumbens (NAc) was chosen due to its relevance in reward learning (Namburi et al., 2015; Sesack and Grace, 2010) and social decision making (Haruno et al., 2014) in conjunction with two dopaminergic midbrain regions, the substantia nigra and the ventral tegmental area (SN/VTA). Additionally, the bed nucleus of the stria terminalis (BST) is considered a part of the extended amygdala complex (Alheid and Heimer, 1988; de Olmos and Heimer, 1999) and could play an antagonistic role in this task, given its involvement in threat encoding mechanisms (Avery et al., 2016; Clauss et al., 2019; Siminski et al., 2020). Finally, the septal area of the basal forebrain could be of particular relevance during trust decisions (Krueger et al., 2007).

Additionally, we revised and extended the methods section (ln. 640):

Volume of interest analyses were performed on the mean timeseries extracted using *nilearn's fit_transform* from anatomical masks from the BLA, CeA (Tyszka and Pauli, 2016), NAc (AAL Atlas), BST (Torrise et al., 2015), SN/VTA (Talairach atlas transformed to MNI space), and basal forebrain (Jülich Brain MPM atlas).

(...)

An additional functional connectivity analysis was performed to verify that the specificity of the BOLD signal is sufficient for distinguishing between the BLA and CeA. To this end, task fMRI data were corrected for white matter and CSF signal and task effects (Ganger et al., 2015) using regression before estimation of the functional connectivity maps of the BLA and CeA seeds.

- was any correction for multiple ROI testing performed?

Following the reviewer's suggestion, we made a clearer distinction between our primary and secondary analyses. We now explicitly state in our *Methods* section that we did not correct for multiple comparison for the NAc, BST, SN/VTA, and basal forebrain VOI analyses, as these were exploratory in nature. However, please note that all of these regions, nevertheless, except the basal forebrain, would survive conservative Bonferroni-correction. As already stated in our manuscript, for the hypothesized BLA and CeA VOIs, a Bonferroni-corrected threshold was used (2 VOIs and 4 conditions). We updated the methods section (ln. 652):

Results of the BLA and CeA VOI analyses were Bonferroni-corrected. Comparisons between learners and non-learners were performed using two-sampled *t*-tests and Spearman correlations of investments (i.e., trust behavior) and subjective trustworthiness ratings. Habituation was investigated using a different model, where each phase and each round was modelled individually (Supplementary Material). Using paired *t*-tests we investigated the differences in the BOLD response between the trustworthy and untrustworthy player in the other volumes of interest based on the percent signal changes as estimated by the GLM. In these secondary, exploratory analyses, no correction for multiple comparison was used. However, note that these regions, except the basal forebrain, would survive conservative Bonferroni-correction.

- authors mention to have run a whole brain analysis, but then results are not reported in the supplemental material. What the authors want to visually inspect?

We confirm that we performed a whole brain analysis as described in our methods section in addition to our VOI analysis, which was the focus of the present submission. As requested, for visual inspection, we provide the whole brain results in our updated supplement:

Supplementary Figure 4. SPM of the group-level whole brain results for the *introduction*, *investment*, *waiting*, and *outcome* phases (rows 1 to 4). Threshold was set to $p < 0.05$ FWE-corrected (voxel-wise, whole-brain).

- authors often report a lack of significance, I would recommend the use of Bayesian statistic before their interpretation

We thank the reviewer for this comment on potentially misrepresenting an absence of evidence as evidence for the absence of an effect. We take this suggestion very seriously but are not sure if an additional analysis is helpful here to learn more about the results. At one point we state that *correlations with investment differences were not significant*, which was the result from traditional, frequentist hypothesis testing. The same applies to our statement that we did not observe a difference between the trustworthy and untrustworthy player in the CeA and BLA. This is the result from the standard SPM/GLM approach, which was our preference as it is well-established in the fMRI literature. However, to better demonstrate that the non-significant effects are indeed negligible, we added a report of the effect sizes and 95% confidence intervals to our results.

- authors mention that each phase of the task was modelled with a separate predictor. Considering how close in time these task phases are, I would like to see how much correlated these predictors are and how much variance is left to be explained by the different phases

As suggested by the reviewer, we extended the *Methods* section to better describe our statistical model. We confirm that Figure 2 depicts the results from the same statistical model (GLM, 4 phases \times 2 players and confounds) applied to the whole brain data (Figure 2B) as well as BLA (Figure 2C) and CeA (Figure 2D) VOI data, which was extracted using anatomical masks.

As requested, we show the results of the correlation analyses between the parameter estimates for the different phases of a trust game trial for each player and the mean of both for the BLA (first row) and CeA (second row):

- there is too little information to evaluate the ROI analysis: some text and figure suggest that a t-test per time point was calculated (and if this is the case they need to explain and show the clusters or point that are significant with appropriate correction), other part of the text suggest they just average the signal per task-phase

In accordance with the reviewer's recommendation, we made our statistical model clearer and easier to understand and updated the manuscript accordingly. We now write in our methods section (ln. 640):

Volume of interest analyses were performed on the mean timeseries extracted using *nilearn's fit_transform* from anatomical masks from the BLA, CeA (Tyszka and Pauli, 2016), NAc (AAL Atlas), BST (Torrissi et al., 2015), SN/VTA (Talairach atlas transformed to MNI space), and basal forebrain (Jülich Brain MPM atlas). To investigate phase-dependent activation in the amygdala subregions, timeseries analyses were conducted based on the estimated percent signal change, using custom python scripts that reproduced SPM's default GLM analysis, using SPM's canonical HRF to convolve the 8 regressors of interest (4 phases \times 2 players), using the realignment parameters as confounds and a DCT-based high-pass filter with SPM's

default $f=1/128$ Hz cut-off frequency to account for signal drifts. Results of the BLA and CeA VOI analyses were Bonferroni-corrected. Comparisons between learners and non-learners were performed using two-sampled *t*-tests and Spearman correlations of investments (i.e., trust behavior) and subjective trustworthiness ratings. Habituation was investigated using a different model, where each phase and each round was modelled individually (Supplementary Material). Using paired *t*-tests we investigated the differences in the BOLD response between the trustworthy and untrustworthy player in the other volumes of interest based on the percent signal changes as estimated by the GLM. In these secondary, exploratory analyses, no correction for multiple comparison was used. However, note that these regions, except the basal forebrain, would survive conservative Bonferroni-correction.

This means, we did neither perform *t*-tests per time point nor average the signal per task-phase. Instead, we analyzed the amplitude of the estimated BOLD response (converted to percent signal change) for the given task phase, which was modeled by the boxcar function convolved with SPM's canonical HRF.

- authors collected measures of trustworthiness, fairness, attractiveness and intelligence. Did they then run multiple regressions? simple correlations for some of the measures, or what exactly?

We confirm that we performed a simple correlation analysis for the subjective ratings. Please note that the focus of this manuscript was the trustworthiness rating and we do not interpret the findings on attractiveness, fairness, and intelligence, which are only reported for completeness.

- in the description of the task, authors seem to use "round" a bit differently as in one case using it with the meaning of "trial" and in another case as "session", this makes it a bit confusing to understand whether the two players were presented in separate sessions and then the order of the session were randomized across participants, or whether rounds with a different player were included in each session and therefore the order of the players randomized also within subjects.

We are sorry for being unclear and confirm that in total there was one session, where one repeated trust game run was performed, which consisted of 20 rounds or trials where the participant played with each of the players. Note, each round comprised four phases, which were relevant for our differential analysis. To make this clearer we explicitly state at the beginning of the results section (ln. 85):

Participants played the repeated trust game inside the MRI scanner, ostensibly with what turned out to be one trustworthy and one untrustworthy player. In reality, the two players were both simulated, with their returns following a pre-programmed response schedule. Within one session and run, in total, participants played 20 rounds with each of the two players (2±20 rounds).

Reviewer #2 (Remarks to the Author):

This is a very interesting and novel study, driven by clear hypotheses about the differential role of amygdala subregions in trust, which were derived from the animal literature and human lesion studies.

It is laudable that the authors used optimised fMRI sequences allowing them to disentangle different subregions of the amygdala.

Overall the methods are sound and the paper is excellently written.

We would like to thank the reviewer for taking the time to thoroughly work through our manuscript and providing these helpful comments. We feel that the reviewer fully understood the scope, methods, and results of our manuscript and are very grateful for the positive evaluation of our submission.

Please see my more detailed comments for further clarification and thought.

Conceptual issues:

Krueger et al 2007 used optimised fMRI sequences in a trust game to examine related questions as here and found the septal region to be associated with unconditional trust. Since that paper's publication, there is been accumulating evidence on sept-hypothalamic and septal region activations being relevant for attachment-related functions, and it would be very interesting how this often forgotten part of the basal forebrain behaves in the context of the authors' task, would the authors be able to discuss this or add a supplementary analysis?

The reviewer makes an interesting comment regarding the potential involvement of the basal forebrain in trust learning. We aimed at keeping the number of explorative analyses limited. However, we agree with the reviewer on its relevance and followed the suggestion to include it in our analysis and added the updated findings to our revised manuscript.

Results, ln. 275:

Finally, an exploratory analysis of the forebrain showed a similar activation difference between the trustworthy and untrustworthy player for learners vs. non-learners (Δ PSC

$\pm 95\%CI = +0.06\pm 0.02$, $U=640$, $p=0.0126$, $d=0.6$, **FIGURE 3G**) that, like the CeA, correlated with Δ *trustworthiness*, $r_s=+0.33$, $p=0.0084$ (**FIGURE 3I**) but not Δ *investment* (**FIGURE 3H**).

..., ln. 308:

When the *outcome* of the player decision was presented, higher activation were observed for the trustworthy compared to the untrustworthy player in the NAc (Δ PSC $\pm 95\%CI = +0.11\pm 0.03$, $t_{61}=6.3$, $p=0.0000$, $d=0.8$), SN/VTA (Δ PSC $\pm 95\%CI = +0.05\pm 0.03$, $t_{61}=3.7$, $p=0.0005$, $d=0.5$), BST (Δ PSC $\pm 95\%CI = +0.08\pm 0.04$, $t_{61}=4.0$, $p=0.0002$, $d=0.5$), and the basal forebrain (Δ PSC $\pm 95\%CI = +0.03\pm 0.03$, $t_{61}=2.1$, $p=0.0361$, $d=0.3$) (**Figure 4**). Moreover, the gain or loss (i.e., *back-transfer - investment amount*) correlated with NAc ($r_s=+0.19$, $p<0.0001$) and BST ($r_s=+0.10$, $p<0.0001$), but this was irrespective of the activation difference between trustworthy and untrustworthy player.

Discussion, ln. 488:

Finally, the observed difference in the septal area of the basal forebrain during *outcome* evaluation could indicate its involvement in updating beliefs about the other players' trustworthiness. This would be in line with previous research that suggested that this region encodes the participant's volatility beliefs (i.e., an estimator for expected uncertainty) in both social and non-social learning tasks (Diaconescu et al., 2017; Iglesias et al., 2013). In the context of our study, we also observed an involvement during the *introduction* phase, showing more activation in learners relative to non-learners for the *untrustworthy* relative to the *trustworthy* player. It has been shown before that the basal forebrain is involved in trust decisions (Krueger et al., 2007) and its differential activity during *outcome* evaluation (higher for the *trustworthy* player) and the *introduction* phase (higher for the *untrustworthy* player in learners) highlights its potentially crucial role in trust learning.

Figures:

FIGURE 3 | Activation differences between untrustworthy and trustworthy player in the preparation phase. BLA activation differences (contrast: untrustworthy - trustworthy) were higher for learners (magenta) vs. non-learners (cyan) (A), correlated with investment differences (B) and post-experiment subjective trustworthiness rating differences (C). The same relationship was found for CeA (D & F), except the correlation with investment differences was not significant (E). An exploratory analysis of the basal forebrain showed a similar relationship (G, H, I).

FIGURE 4 | More activity for the trustworthy (blue) vs. untrustworthy player (orange) during the *outcome* event found in the secondary analyses. A. in the nucleus accumbens (NAc), $t=+6.3$, $p<0.0001$, **B.** the substantia nigra (SN) and ventral tegmental area (VTA), $t=+3.7$, $p=0.0005$, **C.** the bed nucleus of the stria terminalis (BST), $t=+4.00$, $p=0.0002$, and **D.** the septal part of the basal forebrain, $t=2.1$, $p=0.0361$. Thick lines represent the estimated BOLD model.

Statistical issues:

The focus on hypothesis-driven region of interest analyses is laudable and the effects appear robust, I wonder whether it would be important though to divide the results and methods sections into primary and secondary analyses more clearly to

justify that there was no multiple comparison correction for all regions applied? As the main hypothesis of the paper are around two amygdala regions, I think there is no need to adjust them for multiple comparisons, but it would be helpful to clarify the hypothesis-driven, vs more exploratory 2ndary aspect.

The reviewer raises concerns about the number of regions we compared in our analyses, more precisely, that we do not properly distinguish between our primary analysis (amygdala subnuclei) and secondary findings (all other VOI's). We would like to thank the reviewer for this important comment and agree that our presentation does not make this distinction clear enough. We added the following paragraph to the revised introduction (ln. 68):

Our aims were to derive what role the different subnuclei of the amygdala play for different aspects relevant in learning whom to trust, and to link them to neural activation in other subcortical regions that are highly connected with the amygdala. This means, the primary goal of our analyses was to investigate task-dependent BLA and CeA function and how BOLD response changes (a) during the different task-phases that require different cognitive functions, (b) over the course of the experiment where participants learn the task with varying degrees of success, and (c) how it is affected by differences in trust behavior and subjective trustworthiness ratings. The secondary and more explorative goal of our analyses was to determine the involvement of other highly relevant subcortical brain regions (Janak and Tye, 2015). More specifically, nucleus accumbens (NAc) was chosen due to its relevance in reward learning (Namburi et al., 2015; Sesack and Grace, 2010) and social decision making (Haruno et al., 2014) in conjunction with two dopaminergic midbrain regions, the substantia nigra and the ventral tegmental area (SN/VTA). Additionally, the bed nucleus of the stria terminalis (BST) is considered a part of the extended amygdala complex (Alheid and Heimer, 1988; de Olmos and Heimer, 1999) and could play an antagonistic role in this task, given its involvement in threat encoding mechanisms (Avery et al., 2016; Clauss et al., 2019; Siminski et al., 2020). Finally, the septal area of the basal forebrain could be of particular relevance during trust decisions (Krueger et al., 2007).

Minor wording issues:

-The abstract: line 2: "When planning trust behavior," - insert "the" here please

-abstract line 32: replace "believes" with "beliefs"

-I was wondering whether replacing "trustee" with "player" makes more sense in that "trustee" evokes an association of being on the board of a charity or being a trusted person, but of course that is the authors' decision if they think that this is commonly used in other papers, but I find this a bit confusing for the general reader.

-The results section should use the past tense consistently

We are grateful for the reviewer's remarks and revised our manuscript accordingly.

References

- Alheid, G., Heimer, L., 1988. New perspectives in basal forebrain organization of special relevance for neuropsychiatric disorders: the striatopallidal, amygdaloid, and corticopetal components of substantia innominata. *Neuroscience* 27, 1-39.
- Avants, B.B., Tustison, N.J., Song, G., Cook, P.A., Klein, A., Gee, J.C., 2011. A reproducible evaluation of ANTs similarity metric performance in brain image registration. *Neuroimage* 54, 2033-2044.
- Avery, S.N., Clauss, J.A., Blackford, J.U., 2016. The Human BNST: Functional Role in Anxiety and Addiction. *Neuropsychopharmacology* 41, 126-141.
- Calhoun, V.D., Wager, T.D., Krishnan, A., Rosch, K.S., Seymour, K.E., Nebel, M.B., Mostofsky, S.H., Nyalakanai, P., Kiehl, K., 2017. The impact of T1 versus EPI spatial normalization templates for fMRI data analyses. *Hum Brain Mapp* 38, 5331-5342.
- Clauss, J.A., Avery, S.N., Benningfield, M.M., Blackford, J.U., 2019. Social anxiety is associated with BNST response to unpredictability. *Depress Anxiety* 36, 666-675.
- de Olmos, J.S., Heimer, L., 1999. The concepts of the ventral striatopallidal system and extended amygdala. *Ann N Y Acad Sci* 877, 1-32.
- Diaconescu, A.O., Mathys, C., Weber, L.A.E., Kasper, L., Mauer, J., Stephan, K.E., 2017. Hierarchical prediction errors in midbrain and septum during social learning. *Soc Cogn Affect Neurosci* 12, 618-634.
- Ganger, S., Hahn, A., Kublbock, M., Kranz, G.S., Spies, M., Vanicek, T., Seiger, R., Sladky, R., Windischberger, C., Kasper, S., Lanzenberger, R., 2015. Comparison of continuously acquired resting state and extracted analogues from active tasks. *Hum Brain Mapp* 36, 4053-4063.
- Haruno, M., Kimura, M., Frith, C.D., 2014. Activity in the nucleus accumbens and amygdala underlies individual differences in prosocial and individualistic economic choices. *Journal of cognitive neuroscience* 26, 1861-1870.
- Iglesias, S., Mathys, C., Brodersen, K.H., Kasper, L., Piccirelli, M., den Ouden, H.E., Stephan, K.E., 2013. Hierarchical prediction errors in midbrain and basal forebrain during sensory learning. *Neuron* 80, 519-530.
- Janak, P.H., Tye, K.M., 2015. From circuits to behaviour in the amygdala. *Nature* 517, 284-292.
- Krueger, F., McCabe, K., Moll, J., Kriegeskorte, N., Zahn, R., Strenziok, M., Heinecke, A., Grafman, J., 2007. Neural correlates of trust. *Proceedings of the National Academy of Sciences* 104, 20084.
- Namburi, P., Beyeler, A., Yoroazu, S., Calhoon, G.G., Halbert, S.A., Wichmann, R., Holden, S.S., Mertens, K.L., Anahtar, M., Felix-Ortiz, A.C., 2015. A circuit mechanism for differentiating positive and negative associations. *Nature* 520, 675-678.
- Sesack, S.R., Grace, A.A., 2010. Cortico-basal ganglia reward network: microcircuitry. *Neuropsychopharmacology* 35, 27-47.
- Siminski, N., Böhme, S., Zeller, J., Becker, M., Bruchmann, M., Hofmann, D., Breuer, F., Mühlberger, A., Schiele, M., Weber, H., 2020. BNST and amygdala activation to threat: effects of temporal predictability and threat mode. *Behav Brain Res*, 112883.

Sladky, R., Friston, K.J., Trostl, J., Cunnington, R., Moser, E., Windischberger, C., 2011. Slice-timing effects and their correction in functional MRI. *Neuroimage* 58, 588-594.

Torrise, S., O'Connell, K., Davis, A., Reynolds, R., Balderston, N., Fudge, J.L., Grillon, C., Ernst, M., 2015. Resting State Connectivity of the Bed Nucleus of the Stria Terminalis at Ultra-High Field. *Hum Brain Mapp* 36, 4076-4088.

Tyszka, J.M., Pauli, W.M., 2016. In vivo delineation of subdivisions of the human amygdaloid complex in a high-resolution group template. *Hum Brain Mapp* 37, 3979-3998.

REVIEWERS' COMMENTS:

Reviewer #1 (Remarks to the Author):

The author sufficiently addressed my concerns.